# Review of Methods Used for Diagnosing Tuberculosis in Captive and Free-Ranging Non-Bovid Species (2012–2020)

**DOI:** 10.3390/pathogens10050584

**Published:** 2021-05-11

**Authors:** Rebecca Thomas, Mark Chambers

**Affiliations:** 1School of Veterinary Medicine, University of Surrey, VSM Building, Daphne Jackson Road, Guildford, Surrey GU2 7AL, UK; rebecca.thomas22@virginmedia.com; 2School of Biosciences and Medicine, University of Surrey, Edward Jenner Building, Guildford, Surrey GU2 7XH, UK

**Keywords:** MTBC, *Mycobacterium bovis*, immunological, diagnostic, tuberculosis

## Abstract

The *Mycobacterium tuberculosis* complex (MTBC) is a group of bacteria that cause tuberculosis (TB) in diverse hosts, including captive and free-ranging wildlife species. There is significant research interest in developing immunodiagnostic tests for TB that are both rapid and reliable, to underpin disease surveillance and control. The aim of this study was to carry out an updated review of diagnostics for TB in non-bovid species with a focus predominantly on those based on measurement of immunity. A search was carried out to identify relevant papers meeting a pre-defined set of inclusion criteria. Forty-one papers were identified from this search, from which only twenty papers contained data to measure and compare diagnostic performance using diagnostic odds ratio. The diagnostic tests from each study were ranked based on sensitivity, specificity, and diagnostic odds ratio to define high performing tests. High sensitivity and specificity values across a range of species were reported for a new antigenic target, P22 complex, demonstrating it to be a reliable and accurate antigenic target. Since the last review of this kind was undertaken, the immunodiagnosis of TB in meerkats and African wild dogs was reported for the first time. Suid species showed the most consistent immunological responses and highlight a potential dichotomy between humoral and cellular immune responses.

## 1. Introduction

The *Mycobacterium tuberculosis* complex (MTBC) is a group of genetically similar bacteria that cause the disease tuberculosis (TB) in a range of hosts [1]. The MTBC comprises the major pathogenic mycobacteria species *M. tuberculosis, M. bovis, M. africanum*, *M. canettii, M. microti*, *M. caprae, M. pinnipedii*, *M. mungi*, *M. suricattae*, and *M. orygis* [2]. Cattle are considered the primary host of *M. bovis*; however, infection is not limited to livestock but also affects humans and many other free-ranging and captive wildlife species [3]. Notably, the European badger (*Meles meles*) in the United Kingdom, Brushtail possum (*Trichosurus vulpecula*) in New Zealand, and White-tailed deer (*Odocoileus virginianus*) in the United States are all species implicated in transmission of *M. bovis* to livestock [3]. As reviewed by Miller and Olea-Popelka, different control strategies for TB are implemented in different countries based on the level of disease transmission and prevalence within that country, and considering which species are infected or at risk of infection [4]. Common control strategies include surveillance, culling of reservoirs and infected animals, increased biosecurity and vaccination underpinned by diagnostic testing [5].

Zoonotic transmission of TB may be more likely in zoos due to the close contact of staff with animals, as well as the potential for transmission of infection from human to animal, although this is very rare [6]. In addition, zoos may find it problematic to maintain biodiversity and conserve valuable or endangered species and to exchange genetic resources and animals with one another where TB outbreaks occur, as reviewed by Lécu and Ball [7]. Accurate diagnosis of TB in captive wildlife is therefore important but challenging, given the diversity of species susceptible to the MTBC. 

Conventional diagnostic tests for TB are considered the gold-standard and comprise of bacterial culture, histopathology, and post-mortem examination [8]. Often, these conventional tests are used in combination with one another or are used as a confirmatory test for newer immunological diagnostics, as discussed by Salfinger and Pfyffer [9]. Culture and post-mortem examinations are relatively expensive tests that require laboratory facilities for isolation and identification of mycobacteria. Culture is the primary gold-standard test for TB. However, as mycobacteria are slow growing, culture can be protracted as well as being liable to cross-contamination with other environmental bacteria [10]. Culture also varies in sensitivity depending upon the type of sample used [11]. 

Immunological diagnostics based on the humoral immune response rely on the detection of antibodies specific to MTBC antigens. Whilst easy to perform, they may be a poor indicator of TB infection because antibody titers tend to increase as disease progresses [12]. Hence, the humoral response is less reliable for the detection of asymptomatic cases or cases early on in infection but can be used to monitor the progress of infection, as reviewed in Pollock et al., 2001 [13]. Efforts to increase the sensitivity of antibody-based tests have included the use of multiple antigens, most recently, the P22 complex, made up of 118 different antigenic targets, including MPB83, MPB70 and ESAT-6 [14].

In contrast, the cell-mediated immune (CMI) response is characterized by the production of cytokines, such as IFN-γ released by stimulated lymphocytes. As discussed in the Pollock et al. review, relative to antibody production, the CMI response generally occurs earlier after infection and is considered to play a major role in controlling TB [13]. The intradermal delayed-type hypersensitivity tuberculin skin test (TST) involves intradermal injection of tuberculin, a complex mix of antigens derived from *M. bovis*‑purified protein derivative (PPD) and measurement of swelling at the injection site usually 72 hrs later [15]. The TST is generally unreliable in most non-bovine species, such as European badgers, producing a weak response, which can be altered by the stress of capture [15,16]. In addition, the TST is often considered impractical for free-ranging wildlife because of the need to capture and retain the animal to read the test, as discussed and reviewed by De Lisle et al. [15]. 

Indicators often used to measure diagnostic test performance include, but are not limited to, sensitivity and specificity, predictive values, likelihood ratios, and receiver operating characteristic curve (ROC curve) analysis. Another, but less used, method is the diagnostic odds ratio (DOR). The DOR is a single indicator of test performance, being a ratio of the odds of a positive result in a diseased individual relative to the odds of a positive result in a non-diseased individual [17]. DOR can range from 0 to infinity, but a value of 1 demonstrates that the test has no discrimination between an individual with and without disease. The higher the DOR value, the better able a test is in discriminating infected from non-infected individuals [17].

The aim of the current project was to perform a review of diagnostics used for the detection of TB in free-ranging and captive wildlife species with a focus predominantly on those based on measurement of immunity, updating on previous reviews published in 2009 and 2013 [18,19]. Different studies identifying new or modified immunological targets and techniques in either known or novel wildlife reservoirs were identified and explored to evaluate the performance of the diagnostic technique and approaches used. The DOR was used to determine the performance of diagnostic tests for TB, not having previously been used for this purpose in animal studies.

## 2. Results

### 2.1. Summary of Reported Techniques and Species since 2012

A total of 41 papers were identified and considered as relevant. Table 1 shows the test employed, test target, MTBC species, the nature of infection (natural or experimental), and the species of animal being observed for these 41 papers.

Using the data from Table 1, the most frequently used diagnostic tests, target antigens, and studied species were identified. First, it was noted that the most common species studied were wild boar (*Sus scrofa*)*,* deer, and badgers (*Meles meles*)*,* with each species appearing in seven individual studies (16.7% each of the total), closely followed by warthogs (*Phacochoerus africanus*)*,* which appeared in six studies (14.3%) (Table 1). The animals of these species were a range of wild and captive animals, with the deer being a mix of red deer (*Cervus elaphus*) and white-tailed deer (*Odocoileus virginianus*). Species which appeared in fewer studies included elephants (*Elephas maximus* and *Loxodonta africana*)*,* lions (*Panthera leo*) and rhinoceros (*Ceratotherium simum*). Meerkats (*Suricata suricatta*) and African wild dog (*Lycaon pictus*) were reported for the first time in this context, each being the focus of one study (Table 1). 

The most common techniques used within the 41 studies were ELISAs and Lateral Flow devices (LFD), consisting of INgezim TB-CROM (Eurofins Technologies Ingenasa, Madrid, Spain), STAT-PAK (Chembio Diagnostic Systems, Inc., Hauppauge, NY, USA), and Dual Path Platform (DPP) VetTB assay (Chembio Diagnostic Systems, Inc.). Other tests included tests of CMI, such as the Interferon-Gamma Release Assay (IGRA) and TST (both the Comparative Intradermal Tuberculin Test (CITT) and the Single Intradermal Tuberculin Test (SITT)) (Table 1). However, it was evident that serological tests were more frequently selected approaches than those based on CMI.

In parallel with the most common techniques used, the most recurrent antigenic targets were revealed. Most tests used the same or similar antigens, or a mixture of recombinant proteins (Table 1). For instance, bPPD, MPB70, MPB83, ESAT-6, and CFP10 were commonly used as either individual targets or mixed as a cocktail of either ESAT-6 and CFP10 or MPB70 and MPB83. The P22 complex protein [14] was used in a range of lateral flow assays or a ‘P22 ELISA’. 

### 2.2. Statistical Analysis

From the 41 papers, 20 contained data with which to carry out statistical analysis, calculating, if not stated, sensitivity, specificity, negative and positive predictive values (NPV, PPV), DOR, and where suitable the corresponding 95% confidence intervals (95% CI). The false negative and false positive (FN, FP), true negative and true positive (TN, TP) rates were calculated and used in a statistical test to create another set of data (Table 2) that could be used to compare diagnostic performance. The DOR was used as a measure of diagnostic performance, being the ratio of the odds of a positive result in a diseased individual relative to the odds of a positive result in a non-diseased individual [17]. The remaining 22 papers were evaluated, but no statistical analysis was conducted because the appropriate information was missing from the paper, such as true infection status. 

### 2.3. Analyzing Diagnostic Performance

Diagnostic performance was compared across all tests from the twenty studies in Table 2. Data were ordered to find the top ten best performing and lowest ten performing tests based on sensitivity, specificity, and DOR, individually (Table 3 and Table 4). The 95% CI overlapped for nearly all diagnostic tests as many DOR calculations had a large 95% CI; therefore, the tests were simply ranked based on the calculated DOR. Fifty-three% of top-ranking tests were carried out in wild boar, with 8/10 (80%) of the DOR top ranking tests used in suid species. The INgezim TB Porcine test used for wild boar had the highest sensitivity (100%), specificity (100%), and DOR (3111) of any test (Table 3). This was followed by the DPP VetTB assay and an in-house ELISA [49] based on antibody recognition of the P22 protein complex, also tested in suid species. The most frequently used antigenic targets in the top-ranking tests were MPB83, MPB70, P22 complex, ESAT-6, and CFP10. In contrast, the lower ranking tests consisted of more CMI diagnostics, such as the TST. Based on the DOR, the worst performing test were an LFD using an IgG cocktail of commercial anti-mouse IgG and anti-rabbit IgG [42] (DOR, 1.1) and both CITT and SITT. Antigenic targets among the more poorly performing tests included bPPD and MPB83, although MPB83 featured infrequently in comparison to bPPD. Additionally, the lowest ranking tests were carried out on deer, badgers, and lions, with a few studies on warthogs and wild boar. Notably, the TB ELISA-VK using bPPD as a target [33] was ranked within the top 10 for sensitivity but within the lowest 10 for specificity and did not appear among either of the rankings according to DOR.

### 2.4. Importance of Gold-Standard Testing and Knowledge of Infection Status

Through critical analysis of the papers examined, it was apparent that the estimated test performance was dependent on whether the diagnostic samples were derived from naturally or experimentally infected animals, and on the definition of infection status (i.e., the gold-standard applied in the study). An example of the importance of the former was the study by Fresco-Taboada et al. [49], in which they tested a series of techniques separately using experimental samples and field samples. As shown in Table 2 and Table 3, tests on the field samples showed greater accuracy in comparison to the experimental samples, ranking higher in the data analysis. With respect to the gold-standard of infection, the study by King et al. [27], which was excluded from analysis due to not using culture as the gold-standard to define true infection status, is illustrative. In that study, the diagnostic performance of three tests (IGRA; STAT-PAK; and qPCR) was assessed. The study did not use a true gold-standard test but instead interchangeably trialled the STAT-PAK and IGRA, as gold-standards together to form one gold standard, and as indiviudal gold standards. When calculating DOR, because there was no measure to identify TN, FN, TP, and TN, a DOR of 1.0 was generated for each test, no matter which ‘gold standard’ method was used, rendering it impossible to determine the true diagnostic value of any of the tests.

## 3. Discussion

This study was intended as a review of the tests available for diagnosing TB in non-bovid species, focusing on immunological tests and highlighting any advances from previous reviews undertaken in 2009 and 2012 [18,19]. Common indicators of diagnostic performance include sensitivity, specificity, PPV, and NPV; however, these factors are insufficient to demonstrate diagnostic performance alone [17]. Sensitivity and specificity indicators are based on a proportion of results showing positive or negative results among diseased or healthy individuals and do not consider cut off values [17]. NPV and PPV are generally not good indicators of diagnostic performance per se as they are dependent on the prevalence of infection and therefore assess diagnostic performance in a context-dependent situation [17,62]. For this study, DOR was chosen as the primary method of evaluating diagnostic performance because it serves as a single measure of test performance independent of disease prevalence [17], making comparisons across studies more straightforward. This is the first study to use DOR to assess diagnostic test performance in animals, although it has been used to assess the performance of TB tests in humans, e.g., [63].

Wild boar, badger, and deer were the most common species used in studies, with 27.5% of studies carried out in suid species (pigs, wild boar, and warthogs). Wild boar, badger and white-tailed deer are all significant maintenance hosts of TB in different countries [3]. Wild boars have been documented across Europe showing marked increase in numbers [64]. Throughout Europe, wild boar are showing higher levels of transmission of TB, without the requirement of livestock to maintain infection in the ecosystem, as reviewed in Gortázar et al., 2012 [65]. This has an impact on the population of wild boar itself but also increases the chance of transmitting the disease to other wildlife [66]. The increased awareness of wild boar as an important vector of animal TB is reflected the increase in the number of papers reporting the use of immunodiagnostics for suids, 14 papers in this report in comparison to only three papers covering a similar span of time in the last review [19]. The performance of diagnostic tests was reported in two new species since the previous reviews: meerkat and African wild dog, both being the focus of one study each. TB in meerkats is similar to that in other mammalian species [67], and their study has shed light on the behaviors and social interactions that may affect transmission of TB within social mammal species [68]. African Wild Dogs are classed as a threatened species that are currently under high pressure of infection which may impact their long-term survival and conservation [50]. A study looked at 21 packs of wild dog in Kruger National Park, where TB is endemic in African buffaloes and found using an IGRA that 20/21 of the packs studied had been sensitized to *M. bovis*, showing an 83% prevalence of infection [50]. Despite these results, the species is currently considered stable but highlights the potential threat that could occur with changes in biological and environmental factors such as habitat availability and reproductive rates [50].

Antigenic targets identified frequently in this study were ESAT-6, CFP10, MPB83 and MPB70. Recombinant proteins like CFP10/ESAT-6 have demonstrated high sensitivity and specificity for TB detection in people in comparison to conventional CMI diagnostics like the TST [69]. CFP10 and ESAT-6 may also be the target of strong antibody-positive responses when included in serology tests for both elephants and wild boar [52,54] but show poor diagnostic performance in badgers, with no significant increase in antibody response associated with disease progression [70]. Therefore, the diagnostic performance of CFP10 and ESAT-6 antigens cannot be generalized across species, as is the case with many antigenic targets, but does demonstrate potential for accurate detection of TB in certain species. Individually, MPB83 induces high antibody responses across a range of species including cattle, badger, deer, wild boar, and primates [49,71,72,73]. P22 was described in 2017 [14], and therefore, was not reviewed previously. P22 complex is a mix of 118 different proteins, some of the most abundant being MPB70, MPB83, and ESAT-6 [14]. P22 complex was reported to have reduced cross-reactivity with *Mycobacterium avium*, having greater sensitivity than other antigenic targets, like bPPD, [14] in different species, including llamas, cattle, goats, pigs, and sheep [74,75]. In our review, although MPB83 and P22 appeared most frequently as antigenic targets in the top-ranking tests according to sensitivity, specificity, or DOR, they did not appear any more frequently than would be expected by chance, their appearance among the best performing tests more likely indicating how commonly these antigens are used. Nonetheless, both antigens gave good performance in a variety of test platforms against a range of non-bovid species. P22 as an antigenic target gave sensitivity and specificity values of 70.1–96.7% and 75.0–100.0%, respectively, across studies in wild boar, pig, deer, and badgers. Interestingly, the inclusion of multiple antigens usually increases the likelihood of FP occurring, but this was not seen with P22, despite it being a complex of 118 different antigens. When a P22-based ELISA was compared to the diagnostic performance of MPB83 as a target, it produced similar diagnostic results; however, when used in parallel, sensitivity was increased [76]; some infected animals were only detectable using MPB83 antigen, whilst others were only detectable using the P22 complex [76]. This was surprising since MPB83 is an abundant component of P22. Consequently, when used in parallel, a greater range of animal species were detected. More research is required using field samples to compare and validate the potential of P22 across a wider array of species to confirm the findings above.

Serological diagnostics were more common in the present study than CMI tests, with more serological tests appearing in the top ten. Generally, CMI tests are considered to give high sensitivity; however, this was not seen in this review as CMI tests did not appear among the tests with the highest DOR values. In general, the CMI tests were not carried out in suid species but instead in lions and deer, and this could explain the cause of their lower apparent performance, particularly as the high performing tests were carried out in suid species. Suid species are noted to have a detectable humoral response soon after *M. bovis* exposure which is maintained with disease progression, allowing for rapid detection [54,77]. Moreover, as reviewed by Berger, in most species, the humoral antibody response is dependent upon the cell-mediated response initiating a T helper cell response to activate macrophages and other essential cytokines for antibody activation [78]. However, it has been suggested that suid species have a dichotomy between the humoral and CMI response, meaning that a strong humoral response can occur independently of a cell-mediated response [79,80,81]. 

Despite a test having a high accuracy, it did not always correlate with high diagnostic performance, based on DOR. For example, TB ELISA-VK [37], t-bPPD [37], and bPPD2 [37] were all ranked among the top ten performing tests according to DOR but did not appear in the top ten for either sensitivity or specificity. Conversely, the Ingezim TB-CROM [49], Indirect PPD ELISA [33], and TB ELISA-VK [33] appeared in either or both top ten for specificity and sensitivity but not DOR. We reason that DOR is a better metric for assessing the performance of a diagnostic test since sensitivity and specificity (as pooled or indiviudal indicators) do not represent discriminatory performance, since a high sensitivity can be accompanied by a low specificity, as shown particularly for the TB ELISA-VK [33]. In contrast, DOR is a combination of both sensitivity and specificity, increasing when they become near perfect. 

## 4. Materials and Methods

### 4.1. Literature Search and Exclusion Criteria

Using NCBI PubMed, we identified appropriate papers written in English from 2012, when the last review [19] was carried out. A total of 162 papers were found using the search criteria: ((((wild*) AND (mycobacteri*)) AND (diagnos*)) AND ((“2012/09/01”[Date‑Publication]: “3000”[Date‑Publication]))) AND (immun*). For each of the 162 papers, the abstracts were reviewed, looking for details of the use of (immuno)diagnostics for MTBC infection in non-bovid species. Papers were excluded if they were exclusively based on bovid species, mycobacteria that do not cause TB infection such as *Mycobacterium avium* subspecies *paratuberculosis* or used exclusively non-immunological based diagnostic tests, with the exception of Stewart et al. [42] as it reported a novel immunochromatographic lateral flow assay specific for *Mycobacterium bovis* cells. Additionally, previous review articles were excluded from data collection and statistical analysis but were recorded and reviewed for completeness.

From this, forty-one papers were recorded as relevant from which data were collected, including the species under study, whether TB was experimentally or naturally induced, the mycobacterium species, the test used and how it was employed, the target of the test (i.e., antigen(s)), sample size and type of sample, relative sensitivity and specificity of the diagnostic technique, the NPV and PPV, and the associated cut-off values. If any of the information was not present or had not been mentioned, this was noted. 

### 4.2. Statistical Analysis

Using the sample size and infection status, TP and TN, and FP and FN values were calculated from the 41 papers where possible, if not already stated. Studies with missing values, e.g., for sensitivity, specificity, PPV, and NPV, were calculated where possible from the reported data using the following formulae:sensitivity = TP/(TP + FN)(1)
specificity = TN/(TN + FP)(2)
NPV = TN/(TN + FN)(3)
PPV = TP/(TP + FP)(4)

Following this, the 95% CI surrounding the sensitivity and specificity were noted, if available, or calculated if not. The DOR for each individual test was calculated using the formula:DOR = (TP/FN)/(FP/TN)(5)

The DOR was then adjusted, by adding 0.5 to each of the cells in the contingency table, to account for the tests that had ‘0′ values in any of the TN, TP, FP, FN values. The DOR adjustment was applied across all studies to prevent introducing bias to the data. The adjusted DOR was then used to calculate the 95% CI using the formulae below. All calculations were rounded to 1 decimal place. All formulae for the calculations outlined were sourced from [17].
(6)$$\mathrm{Standard}\;\mathrm{Error}\;\left(\mathrm{SE}\right)\;\left(\mathrm{lnDOR}\right)=\sqrt{\frac{1}{TP}}+\frac{1}{FN}+\frac{1}{FP}+\frac{1}{TN},$$
95% CI = lnDOR ± 1.96 × SE (lnDOR)(7)
True 95% CI = ‘=EXP(±CI)’(8)

The most common species, techniques, and antigenic targets were noted, and the data used to rank the tests in order of sensitivity, specificity, and DOR. All study data included in statistical analysis involving test sensitivity and specificity were established using culture as a gold-standard to confirm infection status. 

## 5. Conclusions

In conclusion, a variety of diagnostic tests are now available for an array of wildlife species, with increasing variety of species being studied. The focus of this review was on diagnostic tests that detect or measure the host immune response to infection. From the current review, it was evident that serological tests are surpassing tests like the TST and even other CMI-based tests, such as IGRA for diagnostic performance. Obtaining proof of high accuracy in tests is still an issue, restricting validation of many tests. The current review used DOR to evaluate diagnostic performance, which to the best of our knowledge has not been used previously for assessing TB diagnostic tests in animals. P22 complex was identified as a promising, new antigenic target, which alongside MPB83 demonstrated potential for use as an accurate seroantigenic target. We believe these conclusions to be consistent with the evidence and arguments presented.

## Figures and Tables

**Table 1 pathogens-10-00584-t001:** Summary of forty-one relevant results from a PubMed search identifying papers from 2012 to present, looking at diagnostic tests for TB in free-ranging and captive non-bovid wildlife.

Mycobacterium Species	Natural (N) or Experimental (E) Infection ^1^	Species	Technique Employed	Target	Reference
*M. tuberculosis* and *M. bovis*	N (333)	Deer (*Cervus unicolor swinhoei* and *C. nippon taiouanus*)	Culture, mnPCR, SITT, Acid fast stain	NA	[20]
*M. bovis*	N (483) + E (31)	White-Tailed Deer (*Odocoileus virginianus*)	DPP VetTB Assay	MPB83, CFP10, ESAT-6	[21]
*M. bovis*	N (75)	White Rhinoceros (*Ceratotherium simum*)	IGRA	bPPD, aPPD	[22]
MTBC	N (2080)	Wild boar (*Sus scrofa*)	bPPD ELISA	NA	[23]
*M. bovis*	N (5) + E (15)	Red Deer (*Cervus elaphus*)	EVELISA	NA	[24]
*M. bovis*	N (7) + E (9)	White-Tailed Deer (*Odocoileus virginianus*)	EVELISA	MPB83	[25]
*M. bovis*	N (126)	Wild Boar (*Sus scrofa*)	DPP VetTB, ELISA IgG, ELISA DR, ELISA IgM, Culture	MPB70, MPB83, CFP10, ESAT-6, bPPD, IgG, IgM	[26]
*M. bovis*	N (ND)	European Badger (*Meles meles*)	STAT-PAK, IGRA, qPCR, Culture	MPB83, ESAT-6, CFP10, bPPD, aPPD	[27]
*M. tuberculosis*	N (1)	Black Rhinoceros (*Diceros bicornis*)	STAT-PAK, DPP VetTB, MAPIA	MPB83, ESAT-6, CFP10	[28]
*M. bovis*	N (751)	European Badger (*Meles meles*)	IGRA, STAT-PAK Assays	bPPD, aPPD, MPB83, ESAT-6, CFP10, MPB70	[29]
*M. tuberculosis*	N (5)	Asian elephant (*Elephas maximus*)	DPP VetTB Assay, STAT-PAK	MPB83, ESAT-6, CFP10	[30]
*M. suricattae*	N (111)	Meerkat (*Suricata suricatta*)	Cytokine Release Assay	PC-HP peptide pool	[31]
*M. bovis*	N (3)	Warthog (*Phacochoerus africanus*)	DPP VetTB Assay	MPB83, CFP10, ESAT-6	[32]
*M. bovis*	N (35)	Warthog (*Phacochoerus africanus*)	Indirect PPD ELISA, TB ELISA-VK, DPP VetTB Assay	bPPD, MPB83, ESAT-6, CFP10	[33]
*M. tuberculosis*	N (ND)	Asian Elephant (*Loxodonta africana*)	STAT-PAK, DPP VetTB Assay	MPB83, ESAT-6, CFP10	[34]
*M. bovis*	N (474)	Red Deer (*Cervus elaphus*)	STAT-PAK, DPP VetTB, MAPIA	MPB83, ESAT-6, CFP10, bPPD, MPB70	[35]
*M. bovis*	N (550)	European Badger (*Meles meles*)	IGRA, STAT-PAK, Culture	bPPD, aPPD, MPB83, ESAT-6, CFP10	[36]
MTBC	N (217)	Domestic Pig ^2^ (*Sus scrofa domesticus*)	bPPD ELISA, INgezim TB Porcine, INgezim TB-CROM	MPB70, MPB83, bPPD	[37]
MTBC	N (173)	European Badger (*Meles meles*)	IgG ELISA	MPB83, Rv2873	[38]
*M. bovis*	N (14)	African Lion (*Panthera leo*)	qPCR	MIG/CXCL9, ESAT-6, CFP-10	[39]
*M. bovis*	E (3)	White Rhinoceros (*Ceratotherium simum*)	PPD ELISA, TB STAT-PAK, DPP VetTB Assay	bPPD, aPPD, MPB83, ESAT-6, CFP10, MPB70	[40]
MTBC	N (35)	African elephant (*Loxodonta africana*)	Elephant TB STAT-PAK, DPP VetTB Assay	MPB83, ESAT-6, CFP10	[41]
*M. bovis*	N (541)	European Badger (*Meles meles*)	IMS LFA, qPCR, Culture	*M. bovis* whole cells	[42]
*M. bovis*	N (88)	Warthog (*Phacochoerus africanus*)	Cytokine Release Assay	ESAT-6, CFP-10, TB7.7 peptides	[43]
*M. bovis*	N (170)	Warthog (*Phacochoerus africanus*)	Indirect PPD ELISA, TB ELISA-VK	bPPD	[44]
*M. bovis*	N (34)	Warthog (*Phacochoerus africanus*)	SITT, CITT	bPPD, aPPD	[45]
*M. bovis*	N (678)	Wild Boar (*Sus scrofa*)	bPPD ELISA	bPPD	[46]
*M. bovis*	N (131) + E (2)	White Rhinoceros (*Ceratotherium simum*)	IGRA	ESAT-6, CFP10	[47]
*M. bovis*	E (ND)	European Badger (*Meles meles*)	MPB83-IgA ELISA	MPB83 specific-IgA	[48]
*M. bovis*	N (55) + E (51)	Wild Boar (*Sus scrofa*)	INgezim TB CROM (LFA), INgezim TB Porcine and INgezim Tuberculosis DR, Indirect ELISA	MPB83, MPB70, P22 complex	[49]
*M. bovis*	N (40)	African wild dog (*Lycaon pictus*)	IGRA	ESAT-6, CFP10	[50]
MTBC	N (85) + E (36)	European Badger (*Meles meles*)	P22 ELISA	P22 complex	[51]
MTBC	N (222)	African Elephant (*Loxodonta africana*)	STAT-PAK Assay, DPP Vet TB Assay	MPB83, ESAT-6, CFP10	[52]
*M. bovis*	N (326)	Lion (*Panthera leo*)	STAT-PAK, DPP Vet TB, SITT	MPB83, ESAT-6, CFP10, bPPD	[53]
*M. bovis*	N (79)	Wild Boar (*Sus scrofa*), Warthog (*Phacochoerus africanus*)	DPP VetTB Assay	IgG, MPB83, CFP10, ESAT-6	[54]
*M. bovis*	N (495)	Wild Boar (*Sus scrofa*)	PCR, IDEXX Ab test, INgezim TB porcine, TB ELISA-VK	bPPD, MPB83, MPB70	[55]
*M. bovis*	N (15)	Warthog (*Phacochoerus africanus*)	GEA	ESAT-6, CFP10	[56]
MTBC	N (277)	Wild Boar (*Sus scrofa*)	P22 ELISA, bPPD ELISA	P22 complex, bPPD	[57]
MTBC	N (221)	Red Deer (*Cervus elaphus*)	P22 ELISA, bPPD ELISA	P22 complex, bPPD	[58]
MTBC	N (88)	Red Deer (*Cervus elaphus*)	CITT, Serum Hp	Hp	[59]
*M. bovis*	N (62)	African Lion (*Panthera leo*)	CITT, GEA	bPPD, aPPD	[60]

^1^ Number in parenthesis indicates samples size, ND indicates precise number could not be determined from the paper, e.g., specify trapping events rather than individuals. ^2^ Domestic pigs were exceptionally included here as the study was carried out on domestic free-range Iberian pigs reared outdoors and sharing natural resources with other wild and domestic animals, including cattle and wildlife. mn, Multiplex nested; q, real-time; PCR, Polymerase Chain Reaction; SITT, Single Intradermal Tuberculin Test; CITT, Comparative Intradermal Tuberculin Test; DPP, Dual Platform Pathway; PC-HP peptide pool, contains ESAT-6 and CFP¬10 peptides and antigens derived from the gene Rv3615c and an additional three genes [61]; IGRA, Interferon-Gamma Release Assay; ELISA, Enzyme-linked Immunosorbent Assay; EVELISA, Ethanol Vortex ELISA; STAT-PAK (Chembio Diagnostic Systems, Inc.)/LFA, Lateral Flow Assay; MAPIA, Multi-antigen Print Immunoassay; (b)PPD, (Bovine) Purified Protein Derivative; IMS, Immunomagnetic Separation; GEA, Gene Expression Assay; Hp, Haptoglobin; NA = Not Applicable, was not indicated in the study.

**Table 2 pathogens-10-00584-t002:** Summary of forty-one relevant results from a PubMed search identifying papers from 2012 to present, looking at diagnostic tests for TB in free-ranging and captive non-bovid wildlife.

Species	Test	NPV	PPV	Sens (%) (95% CI)	Spec (%) (95% CI)	DOR	DOR 95% CI	Reference
Deer	mnPCR	0.5 ^1^	0.7	83.3 (60.0–104.0%)	28.6 (−4.8–62.0%)	1.9	0.2–14.6	[20]
	Acid-fast Stain	0.6	1.0	66.7 (49.0–93.3%)	100.0 (100.0%)	28.3	1.3–618.0	
Deer	DPP VetTB (Experimental)	0.8	0.9	58.1 (39.3–74.9%)	98.4 (90.3–99.9%)	57.1	9.8–333.5	[21]
	DPP VetTB (Natural)	0.9	0.7	71.9 (53.0–85.6%)	98.22 (96.4–99.2%)	129.1	46.8–355.8	
	DPP VetTB (Combined)	0.9	0.7	65.1 (51.9–76.4%)	97.8 (96.5–98.6%)	79.5	40.5–156.1	
Deer	EVELISA	0.9	0.9	86.7 (70.0–103.0%)	93.3 (80.7–105.0%)	52.2	6.0–450.7	[24]
Deer	EVELISA	0.9	1.0	87.5 (71.0–103.0%)	100.0 (100%)	295.8	13.3–6593.1	[25]
Warthog	Indirect PPD ELISA	0.9	0.9	87.5 (62.0–98.0%)	89.5 (67.0–99.0%)	40.6	6.2–267.7	[33]
	TB ELISA-VK	0.9	0.8	87.5 (62.0–98.0%)	78.9 (54.0–94.0%)	20.0	3.6–109.8	
	DPP VetTB	0.8	0.9	75.0 (49.0–93.0%)	89.5 (67.0–99.0%)	19.4	3.5–107.3	
Pig	INgezim TB Porcine	0.9	1.0	78.0 (65.3–87.7%)	100.0 (95.9-100.0%)	609.7	35.4–10486.8	[37]
	INgezim TB-CROM	0.9	0.9	74.6 (61.6–85.0%)	98.9 (93.8–100.0%)	167.5	30.2–930.0	
	TB ELISA-VK	0.8	1.0	72.9 (59.7–83.6%)	100.0 (93.8–100.0 %)	466.6	27.3–7961.8	
	t-bPPD ELISA	0.8	1.0	71.2 (57.9–82.2%)	100.0 (95.9–100.0%)	429.9	25.2–7319.7	
	In-house ELISA	0.8	1.0	66.1 (52.6–77.9%)	100.0 (95.9–100%)	341.0	20.1–5781.6	
Wild Boar	MPB83 IgG ELISA	0.9	1.0	86.4 (72.0–100.0%)	100.0 (100.0%)	484.7	23.9–9843.4	[38]
Badger	LFD	0.5	0.5	8.1 (2.7–17.8%)	92.6 (83.7–97.6%)	1.1	0.3–3.8	[42]
	PCR	0.6	0.6	58.1 (44.9–70.5%)	70.6 (58.3–81.1%)	3.3	1.6–6.7	
Warthog	IP-10 Assay	0.9	0.6	68.4 (46.0–85.0%)	83.7 (71.0–91.0%)	10.1	3.1–33.4	[43]
Warthog	SITT	0.8	1.0	68.8 (41.0–89.0%)	100.0 (81.0–100.0%)	77.4	3.9–1534.1	[45]
	CITT	0.9	1.0	81.3 (54.0–96.0%)	100.0 (81.0–100.0%)	142.7	6.8–2998.9	
Rhinoceros	IGRA	0.9	0.8	78.4 (52.3–93.5%)	92.2 (63.9–99.8%)	38.6	13.7–108.8	[47]
Wild Boar	INgezim TB CROM Ab (Experimental)	0.6	1.0	90.2 (78.6–96.7%)	100.0 (66.2–100.0%)	160.6	8.2–3156.4	[49]
	INgezim TB Porcine (Experimental)	0.7	1.0	92.2 (81.1–97.8%)	100.0 (66.2–100.0%)	200.6	9.9–4043.2	
	INgezim Tuberculosis DR (Experimental)	0.6	1.0	86.3 (73.7–94.3%)	100.0 (66.2–100.0%)	112.7	5.9–2147.9	
	In-house ELISA (Experimental)	0.5	1.0	84.3 (71.4–93.0%)	100.0 (66.2–100.0%)	97.2	5.2–1834.4	
	INgezim TB CROM Ab (Field)	0.9	0.9	93.3 (77.9–99.0%)	96.0 (79.6–99.3%)	186.2	22.9–1513.1	
	INgezim TB Porcine (Field)	1.0	1.0	100.0 (88.3–100.0%)	100.0 (86.2–100.0%)	3111.0	59.6–162400.9	
	INgezim Tuberculosis DR (Field)	0.9	1.0	93.3 (77.9–99.0%)	100.0 (86.2–100.0%)	581.4	26.6–12689.2	
	In-house ELISA (Field)	0.9	1.0	96.7 (82.7–99.4%)	100.0 (86.2–100.0%)	1003	39.1–25719.3	
Badger	Indirect ELISA	0.9	0.7	81.4 (71.4–91.3)	75.0 (66.3–83.6%)	12.5	5.7–27.5	[51]
	Competitive ELISA	0.9	0.8	78.0 (67.0–88.0%)	89.6 (83.0–95.6%)	28.4	11.7–68.5	
Lion	STAT-PAK	0.8	1.0	62.5 (35.0–85.0%)	100.0 (78.0–100.0%)	69.5	3.6–1353.5	[53]
	SITT	0.8	0.7	72.7 (39.0–94.0%)	80.0 (52.0–96.0%)	8.7	1.6–48.4	
Warthog, Wild Boar	DPP VetTB Assay (Wild Boar)	0.7	0.9	80.4 (68.0–88.0%)	96.7 (81.9–100.0%)	77.8	13.3–453.8	[54]
	DPP VetTB Assay (Warthog)	0.9	0.9	82.6 (62.3–93.6%)	91.4 (76.9–97.8%)	40.2	8.9–181.3	
Wild Boar	PCR	0.9	0.3	62.5 (24.6–91.5%)	97.1 (94.8–98.5%)	50.2	11.6–216.9	[55]
	TB ELISA-VK (0.2 Cut-off)	1.0	0.1	85.7 (42.1–99.6%)	87.5 (83.7–90.6%)	30.0	4.9–181.5	
	TB ELISA-VK (0.5 Cut-off)	0.9	0.4	85.7 (42.1–99.6%)	97.3 (95.2–98.7%)	150.8	23.1–987.0	
	IDEXX	1.0	0.3	75 (34.9–96.8%)	96.7 (94.7–98.1%)	74.1	15.9–345.5	
	INgezim TB Porcine	0.9	0.3	75 (34.9–96.8%)	96.9 (94.9–98.3%)	79.1	16.9–370.4	
Warthog	GEA (CXCL9)	0.7	0.9	60.0 (32.0–84.0%)	94.1 (71.0–100.0%)	16.1	2.3–112.6	[56]
	GEA (CXCL10)	0.9	0.9	86.7 (60.0–98.0%)	94.1 (71.0–100.0%)	59.4	6.9–509.1	
	GEA (CXCL11)	0.7	1.0	53.3 (27.0–79.0%)	100.0 (80.0–100.0%)	39.7	2.0–779.2	
	GEA (TNF-α)	0.8	0.8	73.3 (45.0–92.0%)	88.2 (64.0–99.0%)	15.8	2.8–88.8	
	GEA (IFN-γ)	0.8	0.8	80.0 (52.0–96.0%)	82.4 (57.0–96.0%)	14.8	2.8–78.1	
Wild Boar, Pig	P22 ELISA	0.9	0.9	84.1 (79.3–98.4%)	98.4 (96.5–99.4%)	291.0	125.7–673.6	[57]
	bPPD ELISA	0.8	0.9	77.3 (71.9–82.1%)	97.3 (95.0–98.3%)	114.7	58.4–225.2	
Deer	P22 ELISA	0.8	1.0	70.1 (63.6–76.0%)	99.0 (96.5–99.8%)	189.4	52.7–681.1	[58]
	bPPD ELISA	0.7	0.9	70.1 (63.6–76.0%)	91.6 (86.9–95.0%)	25.1	14.2–44.2	
Deer	CITT	0.8	0.5	25.0 (−5.0–55.0%)	92.0 (81.3–102.6%)	3.6	0.5–25.6	[59]
	Hp	0.9	0.7	62.5 (28.9–96.0%)	92.0 (81.0–102.0%)	14.8	2.3–95.8	

**^1^** Shaded cells indicate values that were calculated by us and not provided in the original study.

**Table 3 pathogens-10-00584-t003:** The ten best performing tests in non-bovid wildlife ranked based on highest values for sensitivity, specificity, and diagnostic odds ratio (DOR).

Ranking	Top 10 Tests—Sensitivity			Top 10 Tests—Specificity			Top 10 Tests—DOR		
Position	Test	Species	Ag	Test	Species	Ag	Test	Species	Ag
1	INgezim TB Porcine (F ^1^) [49]	Wild Boar	MPB83/ 70	INgezim TB Porcine (F) [49]	Wild Boar	MPB83/ 70	INgezim TB Porcine (F) [49]	Wild Boar	MPB83/70
2	In-house ELISA (F) [49]	Wild Boar	P22 complex	In-house ELISA (F) [49]	Wild Boar	P22 complex	DPP VetTB Assay [54]	Pig	MPB83/ESAT-6/CFP10
3	DPP VetTB Assay [54]	Pig	MPB83/ESAT-6/CFP10	DPP VetTB Assay [54]	Pig	MPB83/ ESAT-6/ CFP10	In-house ELISA (F) [49]	Wild Boar	P22 complex
4	INgezim TB-CROM (F) [49]	Wild Boar	MPB83	INgezim Tuberculosis DR (F) [49]	Wild Boar	MPB83	INgezim TB Porcine [37]	Pig	MPB83/70
5	INgezim Tuberculosis DR (F) [49]	Wild Boar	MPB83	INgezim TB Porcine (E) [49]	Wild Boar	MPB83/ 70	INgezim Tuberculosis DR (F) [49]	Wild Boar	MP83
6	INgezim TB Porcine (E ^2^) [49]	Wild Boar	MPB83/ 70	INgezim TB-CROM (E) [49]	Wild Boar	MPB83	MPB83 IgG ELISA [38]	Badger	MPB83
7	INgezim TB-CROM (E) [49]	Wild Boar	MPB83	EVELISA [25]	Deer	MPB83	TB ELISA-VK [37]	Pig	bPPD
8	EVELISA [25]	Deer	MPB83	MPB83 IgG ELISA [38]	Badger	MPB83	t-bPPD In-house ELISA [37]	Pig	Treated bPPD
9	Indirect PPD ELISA [33]	Warthog	bPPD	INgezim Tuberculosis DR (E) [49]	Wild Boar	MPB83	bPPD2 In-house ELISA [37]	Pig	bPPD
10	TB ELISA-VK [33]	Warthog	bPPD	In-house ELISA (E) [49]	Wild Boar	P22 complex	EVELISA [25]	Deer	MPB83

^1^ F = field samples. ^2^ E = experimental samples.

**Table 4 pathogens-10-00584-t004:** The ten lowest performing tests in non-bovid wildlife ranked based on lowest values for sensitivity, specificity, and diagnostic odds ratio (DOR), with position ‘1’ being the lowest value out of the ten tests.

Ranking	Lowest 10 Tests—Sensitivity			Lowest 10 Tests—Specificity			Lowest 10 Tests—DOR		
Position	Test	Species	Ag	Test	Species	Ag	Test	Species	Ag
1	LFD [42]	Badger	*M. bovis* whole cells	mnPCR [20]	Deer	NA ^1^	LFD [42]	Badger	*M. bovis* whole cells
2	CITT [59]	Deer	a/b PPD	PCR [42]	Badger	*M. bovis* whole cells	mnPCR [20]	Deer	NA
3	GEA CXCL11 [56]	Warthog	ESAT-6/CFP10	Indirect ELISA [51]	Badger	P22 complex	PCR [42]	Badger	*M. bovis* whole cells
4	DPP VetTB assay [21]	Deer	MPB83/CFP10/ESAT-6	TB ELISA-VK [33]	Warthog	bPPD	CITT [59]	Deer	a/b PPD
5	PCR [42]	Badger	*M. bovis* whole cells	SITT [53]	Lion	bPPD	SITT [53]	Lion	bPPD
6	GEA CXCL9 [56]	Warthog	ESAT-6/CFP10	GEA IFNγ [56]	Warthog	ESAT-6/ CFP10	IP-10 assay [43]	Warthog	ESAT-6, CFP-10, TB7.7 peptides
7	Phase Range Serum Hp [59]	Deer	Hp	IP-10 assay [43]	Warthog	ESAT-6, CFP-10, TB7.7 peptides	Indirect ELISA [51]	Badger	P22 complex
8	PCR [55]	Wild Boar	NA	TB ELISA-VK [55]	Wild Boar	bPPD	Phase Range Serum Hp [59]	Deer	Hp
9	STAT-PAK [53]	Lion	NA	GEA TNFα [56]	Warthog	ESAT-6/CFP10	GEA IFNγ [56]	Warthog	ESAT-6/CFP10
10	DPP VetTB assay [21]	Deer	MPB83/CFP10/ESAT-6	DPP VetTB assay [33]	Warthog	MPB83/ESAT-6/CFP10	GEA TNFα [56]	Warthog	ESAT-6/CFP10

^1^ NA = Not Applicable, was not indicated in the study. As P22 complex and MPB83 appeared in 20/51 of the tests studied (39.2%), with 10% of DOR top-ranking tests using P22 complex and 60% using MPB83, we asked whether they were genuinely better antigens of choice, testing the hypothesis that either MPB83 or P22 were over-represented more than expected by chance alone in the best performing tests using the Fisher’s exact test. The *p* values for a two-tailed test showed that neither P22 (*p* = 0.67) nor MPB83 (*p* = 1.08) appeared more often in the top-ranking tests than would be expected by chance. Therefore, there was no evidence to indicate they were superior antigens but simply a reflection of how commonly they were used.

## Data Availability

Data are either contained within the article or relate to 3rd Party Data. Full references are provided for these 3rd Party Data but restrictions may apply to their availability.

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
