# Peer review of "Review of Methods Used for Diagnosing Tuberculosis in Captive and Free-Ranging Non-Bovid Species (2012–2020)"

_pathogens, 2021, doi:10.3390/pathogens10050584_

Round 1

Reviewer 1 Report

The aim of this study was to conduct an updated literature review (2012-2020) on the immunodiagnosis of tuberculosis in species other than cows (wild-life species including). From this search, 42 manuscripts were identified, of which only 20 contained data useful for statistical analysis.

The research topic is very interesting and important from the point of view of zoonoses.

The topic is quite original and brings new information to the subject area compared to other published material. It is a creative analysis of literature data based on a well-done analysis of the results and statistical analysis of the collected data.

The conclusions are consistent with the arguments presented in the text of the manuscript and also relate to the main aim of the study. 

The manuscript is generally very well written and clearly presented.

Author Response

Thank you for your encouraging comments. We don't see any specific issues to address in response to this review.

Reviewer 2 Report

The work presented is very correct and absolutely interesting.

In addition, it is also very useful notice in a practical point and point of view in the field.

However I suggest and ask for a slight implementation of the conclusions. Can you confirm and detail in two additional lines that the Conclusions consistent with the evidence and arguments globally presented ?

Author Response

English is the first language of both authors and we have reviewed the paper for its accuracy.

If understood correctly, we find the suggestion a rather unusual request. However, we have now added the following sentence to the end of the Conclusions:

"We believe these conclusions to be consistent with the evidence and arguments presented."

Reviewer 3 Report

This is a very complete insightful and clear update of the immunological-based diagnostic methods available for diagnosing tuberculosis in non-bovid species. 

I have some minor comments and corrections to suggest:

Table 1, reference 38 deals with wild boar not with badgers as stated in the table

All over but even more readable at lines 251-252, it is MPB83 and P22 based test (not directly the antigen being a test).

Line 283: did not 

In table 2 you include direct diagnosis tests not based in any immunological response, shouldn’t they be excluded ? 

Author Response

This is a very complete insightful and clear update of the immunological-based

diagnostic methods available for diagnosing tuberculosis in non-bovid species.

I have some minor comments and corrections to suggest:

Table 1, reference 38 deals with wild boar not with badgers as stated in the table

We apologise for this mistake that is now corrected.

All over but even more readable at lines 251-252, it is MPB83 and P22 based test

(not directly the antigen being a test).

We have improved the description here at lines 161-62, 258, 262-263, 266-70,

Line 283: did not

Changed. Now at line 290.

In table 2 you include direct diagnosis tests not based in any immunological

response, shouldn’t they be excluded ?

We believe this matter has been dealt with through our response to Reviewer 4.

Reviewer 4 Report

This paper describes the results of a review of the recent (post-2012) veterinary/scientific literature in the field of diagnostic tests for the detection of TB (Mycobacterium tuberculosis complex infections) in free-ranging and captive wild mammals. The authors provide an objective comparison of the performance of these tests using a statistical parameter known as the diagnostic odds ratio (DOR), in addition to the more classic measures of diagnostic test accuracy such as sensitivity, specificity and predictive values.

I consider this review paper is well written, its results are based on a generally sound methodology and it represents a significant scientific contribution to the field of TB diagnostics by pulling together and summarising in a systematic way the results from a vast array of studies of different diagnostic methods conducted in different species and countries.

It can be difficult and somewhat misleading to compare studies of diagnostic methods conducted in different species and families of animals (i.e. comparing apples and oranges). However, I think the authors have done a good job avoiding overinterpretation of their findings and not falling into the trap of giving pooled estimates of diagnostic accuracy when the same test has been used across different species.

Detailed comments:

The title and the last paragraph in the introduction of this paper are somewhat misleading and should be amended, since the authors did include a small number of non-immunological methods for TB diagnosis among the studies selected for review, such as bacteriological culture and qPCR (see for instance refs. 20, 27, 36, 39 and 42 in Table 2).

Abstract (line 22): no need to capitalise "meerkats".

Introduction:

Line 35: ditto, do not capitalise "vulpecula".

Line 55: insert a comma between "growing" and "culture".

Line 65: start a new paragraph from "In contrast".

Line 67: replace 'produced' with 'released', to avoid repetition of this word in the same sentence.

Line 73: insert "most" between "in" and "non-bovine".  For instance, the tuberculin skin test is widely used for ante-mortem TB screening of captive deer species.

Line 76: capitalise "de lisle" (author surname).

Results:

Table 1:

  • It would be helpful if the authors could indicate (e.g. in parentheses) the number of individuals of the relevant animal species sampled/included in each of the 42 studies selected for review, so that readers of this paper can have an idea of the study sizes.  The larger the study size, the more relevant its results and the more robust/precise the diagnostic test accuracy estimates.
  • For some of the studies included in this table, IFN-gamma and IP-10 are listed under the 'Target' column, but those are analytes (cytokines) being measured by the diagnostic assay, rather than the antigen(s) used to stimulate the release of those cytokines by the white blood cells. Should the target antigens not be listed instead?
  • Ref [37] about domestic pigs. Why is this study included in Table 2?  I though that the review was only about diagnostic tests in wild animals, as stated in the last paragraph of the Introduction. Clarify in a table footnote why this reference was included here (domestic pigs reared under extensive management conditions?).

Line 121: I thought that DPP stood for 'dual path platform'.

Table 2:

  • Reference [55]: Domestic pig is not explicitly mentioned in Table 1 in relation to this study (ref no. [55]), but it is mentioned here and in Table 3. Which table is correct?
  • Reference [60]: this gives a negative value for the lower limit of the 95% CI for Se estimate and the upper limit of the 95% CI for the Sp estimates is greater than 100%.  Do these need to be corrected? 
  • In Tables 2 & 3 the authors use "boar", but in Table 1 you wrote "wild boar". Use either term consistently in all the tables and throughout the paper, to avoid confusion.  I prefer "wild boar".

Materials & Methods:

Literature search and exclusion criteria: The authors should clarify whether they only included in the search papers that had been written in English, or not.

Author Response

Please find attached our point by point response to your review, for which we thank you.
